# Genesis of Hawaiian lavas by crystallization of picritic magma in the deep mantle

Junlong Yang [1,2], Chao Wang [1] ✉, Junfeng Zhang [1] & Zhenmin Jin[1]

Olivine is the dominant phenocryst or xenocryst of Hawaiian tholeiitic basalts, and the general consensus is that lavas with MgO concentrations from 7.5 to about 15 weight percent were derived from their primary magmas, which contain ~18–20 weight percent MgO, by only olivine crystallization. However, the major element composition of estimated primary magmas through olivine crystallization correction is inconsistent with direct partial melting of either mantle peridotite or its hybrid with subducted oceanic crust. Our melting experiments on peridotite-derived melt composition show that this discrepancy can be resolved if the primary magmas experienced two other processes before abundant olivine fractionation. First, the primary magmas experienced crystallization of clinopyroxene and garnet in the chamber at the base of the lithosphere (approximately the depths of 90–100 km). Second, the evolved magmas re-equilibrated with harzburgite when passing through the lithospheric mantle (approximately the depths of 60–10 km). Different from the isotopic evidence, the major and rare earth element compositions of Hawaiian post-shield alkali basalts and shield tholeiites suggest that they form from the same source by assimilating different amounts of orthopyroxene.

The petrogenesis of Hawaiian lavas is controversial. Many hypotheses have been proposed to resolve this issue[1–7]. A valid model must match the major and trace element characteristics of Hawaiian lavas[8]. Here, we have evaluated these hypotheses by comparing high-pressure experimental results with compositions of Hawaiian lavas (Supplementary Fig. 1). Because the lithospheric thickness under the plume axis of Hawaii is ~100–110 km (ref. [9]), we use the experimental data with pressures of 3–4 GPa.

The trace element characteristics of Hawaiian lavas, such as the enrichment of incompatible elements and the depletion in elements compatible with garnet, imply that Hawaiian primary magmas were formed by low-degrees (e.g., ~5 wt%) melting of garnet lherzolite[10–12]. However, such melt is too deficient in $FeO^T$ and $SiO_2$ contents and enriched in CaO and $Al_2O_3$ contents to be the parental magma of Hawaiian lavas (Supplementary Fig. 1). Its re-equilibration with harzburgite by assimilating orthopyroxene and precipitating olivine at lithospheric depths can resolve the deficiency of the $SiO_2$ content, but not the differences in the $FeO^T$, $Al_2O_3$, and CaO contents[1,3,13].

Recycled oceanic crust is widely accepted as the main form of heterogeneity in Earth's mantle[14–17]. It is proposed that the parental magma of Hawaiian tholeiitic lavas was produced by mixing the partial melts of garnet lherzolite and silica-saturated pyroxenite formed by the reaction of peridotite with partial melts of recycled oceanic crust[5,6]. This hypothesis explains the high $SiO_2$ and NiO contents and the low CaO content of Hawaiian magmas but not the $Al_2O_3$ and $FeO^T$ contents[5,18–20] (Supplementary Fig. 1).

Many previous studies assumed that the composition of the primary magmas of Hawaiian lavas did not change during transit from the mantle to the crust. However, this may not be accurate[1,3,13]. Here, we present a new high-pressure crystallization model to explain the generation of Hawaiian lavas. Our model is inspired by previous studies on pyroxenite xenoliths of Hawaiian lavas and experiments on material that estimates the primary Kilauea tholeiite composition[3,21–23]. These works provide two important notions. First, the primary magma should have experienced fractionation in the deep lithospheric mantle[21,23]. Second, the estimated parental magmas of Hawaiian lavas

[1]State Key Laboratory of Geological Processes and Mineral Resources, School of Earth Sciences, China University of Geosciences, Wuhan 430074, China. [2]Department of Earth and Space Sciences, Southern University of Science and Technology, Shenzhen 518055, China. ✉e-mail: wangchao@cug.edu.cn

are not in equilibrium with garnet lherzolite on its liquidus at any pressure but are co-saturated with olivine and orthopyroxene at 1.4 GPa and 1425 °C (refs. [1,3]). We suggest that Hawaiian tholeiitic lavas are formed by partial melting of peridotite with subsequent fractional crystallization in the deep magma chamber and re-equilibration with harzburgite in the lithosphere. To test this hypothesis, we experimentally determine the melt composition and melting phase relations of a peridotite-derived melt composition at a pressure of 3.0 GPa.

## Results

### Experimental results

The composition of our experimental starting material is similar to the partial melt of peridotite at 3.0 GPa and 1540 °C (Run #30.14, ref. [24]). The experimental details and run products are fully described in the Methods, Supplementary Figs. 2–4 and Supplementary Table 1. Olivine is present in all runs with a stable proportion of about 3 wt% except the 1350 °C run (#R1244). Orthopyroxene disappears while clinopyroxene and garnet are stable phases in runs with temperatures lower than 1350 °C. The proportions of clinopyroxene and garnet gradually increase with decreasing temperature in these runs. Approximately 2 wt% phlogopite is observed in the 1200 °C run. The melt compositions are plotted against the MgO contents in Supplementary Fig. 1. With decreasing temperature, the $FeO^T$, $TiO_2$, and alkaline contents increase, the MgO and $SiO_2$ contents decrease, and the CaO and $Al_2O_3$ contents first increase due to the crystallization of olivine and orthopyroxene and then slightly decrease due to the formation of garnet and clinopyroxene in the precipitates.

## Discussion

### Migration of primary magma

The permeability of primary magma of ocean island basalts in deep mantle rocks is closely related to the porosity of the latter[25,26]. In the source region, partial melting will result in a porous region facilitating the upward migration of the primary magma[26,27]. However, the lithospheric mantle is relatively cold and pore-free compared to the asthenosphere. Therefore, most primary magma should first accumulate at the lithosphere-asthenosphere boundary (~90–100 km) for some time. This is consistent with the near global observation of a sharp drop in seismic velocity near the base of the lithosphere[28,29]. The migration of magma through the lithosphere is poorly constrained compared to the asthenosphere due to a lack of relevant data. The small grain-size channel may be a viable mechanism for magma migration in the melt zone and the asthenosphere, but highly unlike in the lithosphere. A stable magma flux would require the constant operation of a large amount of small grain-size channels that will destroy the rigidity of the lithosphere. However, lithosphere thinning is not observed in Hawaii. Hence, the most possible mechanism for the migration of magma in the Hawaiian lithosphere is discontinuous conduits or magma fracturing[27,30,31]. Seismic studies indicate that Hawaiian magma starts its explosive ascent at a depth of about 60 km (refs. [32,33]).

During accumulation, heat exchange is inevitable between the hot primary magma and the relatively cold lithospheric mantle[34]. Our experimental results show that this heat exchange will result in the precipitation of clinopyroxene and garnet from the primary magma, which may account for the formation of the garnet pyroxenite xenoliths from Hawaii[23]. Previous experimental results show that peridotite-derived primary magma is formed by the reaction of olivine + clinopyroxene + garnet = melt + orthopyroxene (refs. [24,35]). Therefore, the fractionated magma should be out of equilibrium with orthopyroxene and react with the latter to form olivine + clinopyroxene + garnet. This is consistent with the disappearance of orthopyroxene in our low-temperature experiments and previous experimental results on silica undersaturated compositions[36–39]. The reaction between the evolved primary magma and the orthopyroxene in surrounding peridotite almost stops soon after the formation of a thin layer consisting of

olivine, clinopyroxene, and garnet that isolates the melt from the peridotite. The further reaction requires diffusion of elements (by grain-boundary diffusion or lattice diffusion) through this layer, which is a very sluggish process[40]. Hence, the derivation of primary magma in the deep mantle should be closer to a pure fractionation process. Magmas that are close to lithospheric peridotite would have lower temperatures and higher degrees of crystallization, while the magmas that are relatively far away from lithospheric peridotite would have high temperatures and lower degrees of crystallization. Thus, the magma ascending from the depth of 60 km should be a mixture of magmas that have experienced different degrees of crystallization of clinopyroxene and garnet. Here, we consider two kinds of mixtures (Supplementary Table 1). One is the average of melt compositions with a melt fraction (MF) ranging from 85% to 9%, which corresponds to a total MF of ~40% (AM-40). The other is the average of melt compositions with an MF ranging from 55% to 9%, which corresponds to a total MF of ~28% (AM-28).

Because the primary-phase volume of olivine in basaltic melt expands with decreasing pressure, the evolved primary magma will become undersaturated in orthopyroxene component at pressures lower than about 2 GPa (refs. [3,41]). Therefore, the evolved primary magma will start to react with surrounding peridotite by assimilating orthopyroxene and crystallizing olivine at depths shallower than ~60 km (refs. [42,43]). Many dunites found in ophiolites and other mantle sections were always attributed to such a reaction[43]. The ratio of orthopyroxene assimilated to olivine crystallized for a natural silicate system ranges from 1.2 to 1.4 (ref. [41]). As a consequence of this reaction, the volume of melt and the porosity of harzburgite both increase and thus the magma could rise rapidly through reactive porous flow.

As shown in Fig. 1, the AM-40 and AM-28 have lower $SiO_2$ and higher $FeO^T$, $TiO_2$, CaO, and $Al_2O_3$ contents compared to Hawaiian lavas. The assimilation of orthopyroxene by the AM-40/AM-28 at low pressures will drive their compositions towards the composition trend of Hawaiian lavas. Within experimental error, Hawaiian shield tholeiitic lavas could be derived from the AM-40/AM-28 after the assimilation of about 0–30 wt% orthopyroxene (the solid line with dots in Fig. 1). The orthopyroxene assimilation must be accompanied by olivine precipitation, which is crucial for determining the composition of derivative magma. However, it should be noted that this comparison is made for Hawaiian lavas composed of melt with various amounts of olivine[44]. If the precipitated olivine doesn't separate from the magma, the lava composition will lie on the evolution trend of the AM-40/AM-28. If some of the precipitated olivines separate from or add to the derivative magma, the lava composition will deviate from the evolution trend of the AM-40/AM-28 and move along the olivine addition or subtraction trend (the black dash arrows in Fig. 1).

### High-pressure crystallization model

Our experimental results show that Hawaiian lavas could be generated from mantle peridotite through three steps. In the first step, primary magmas of Hawaiian lavas were formed by partial melting of mantle peridotite at pressures of 3–4 GPa. In the second step, the primary magmas precipitated large amounts of clinopyroxene and garnet due to heat exchange with the surrounding mantle at the base of the lithospheric mantle, which is consistent with recent studies on the basanite-nephelinite glasses of Kilauea[45]. In the last step, these derivative $FeO^T$-rich and $SiO_2$-deficient magma reacted with the wall peridotite by assimilating orthopyroxene and crystallizing olivine in transit through the shallower lithosphere, ultimately forming the Hawaiian shield-building tholeiite lavas.

### Ni in Hawaiian magma and olivine

Figure 2a shows the Ni contents in the primary and parental magmas at a pressure of 3.0 GPa. The calculation method (the same as in ref. [5]) is based on the phase compositions and proportions identified in

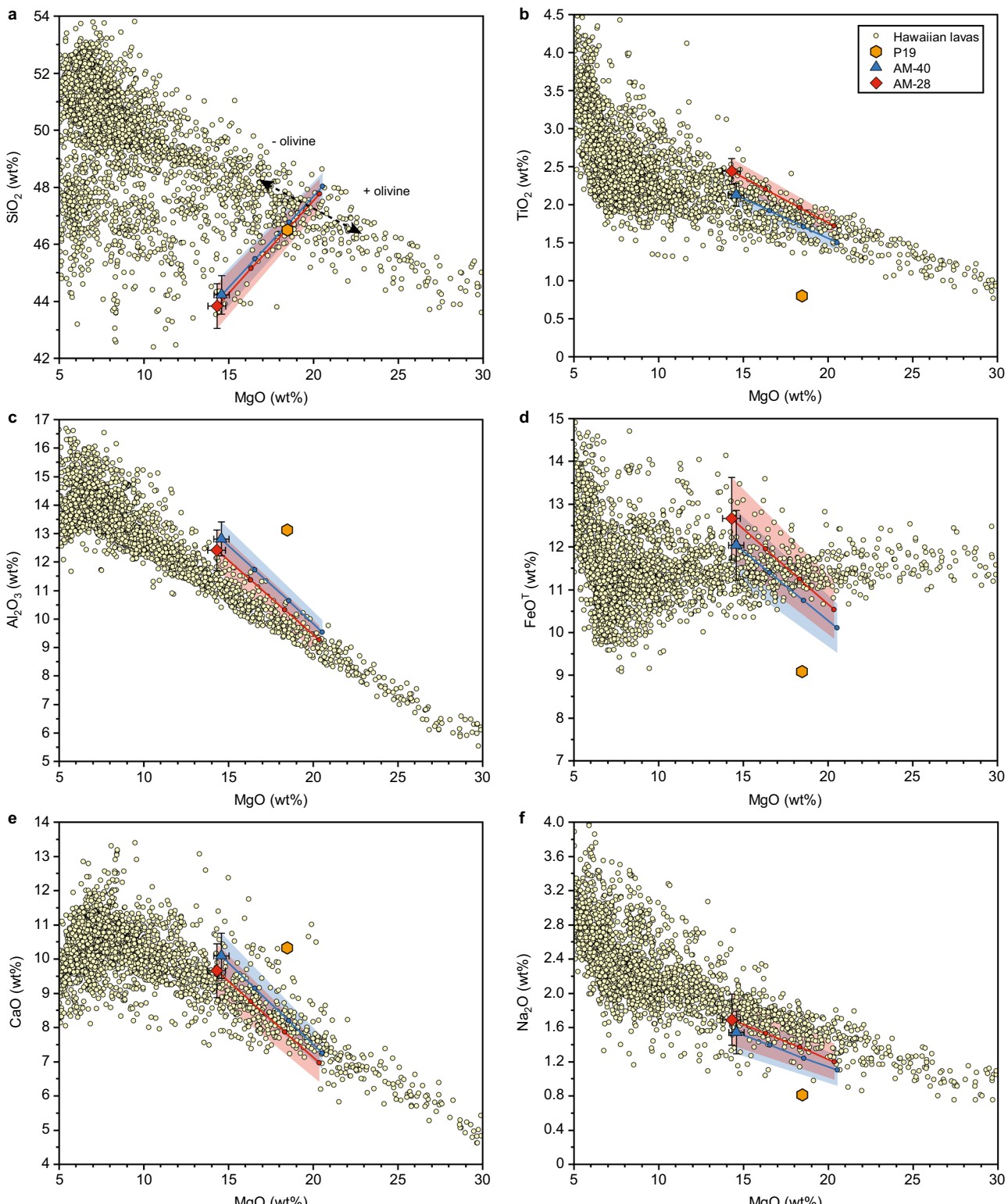

**Fig. 1 | The composition evolution of Hawaiin magma during its ascent towards the surface.** Compositions of peridotite-derived primary melt (P19) and melts derived from it through clinopyroxene and garnet precipitation (AM-40 and AM-28) in the deep magma chamber and subsequent orthopyroxene assimilation (AM- 40 + Opx and AM-28 + Opx) compared to the compositions of Hawaiian lavas in terms of (**a**) $SiO_2$-MgO, (**b**) $TiO_2$-MgO, (**c**) $Al_2O_3$-MgO, (**d**) $FeO^T$-MgO, (**e**) CaO-MgO and (**f**) $Na_2O$-MgO. P19, the starting material used in this study, represents the primary melt formed by the partial melting of peridotite with a melt fraction of about 24 wt% (ref. [24]). The AM-40 and AM-28 are the mixture of melts that have experienced various degrees of clinopyroxene and garnet precipitation (see text and Supplementary Table 1). Each dot on the solid line represents 10 wt% orthopyroxene assimilation. Error bars correspond to one standard deviation. The blue and red shaded areas represent possible ranges of the composition of parental magmas within experimental errors. Black dash arrows indicate the effects of olivine addition and subtraction. Data on Hawaiian lavas are from the GEOROC database.

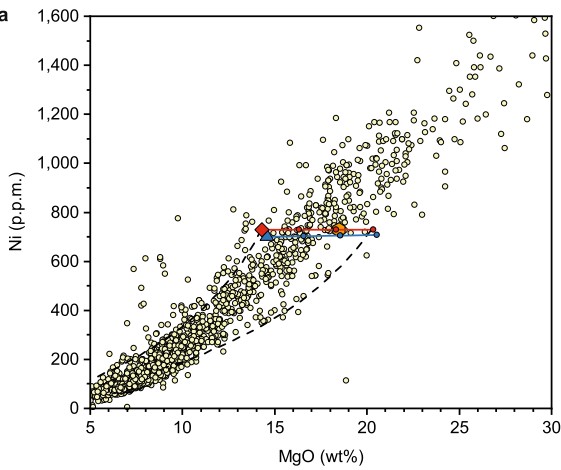
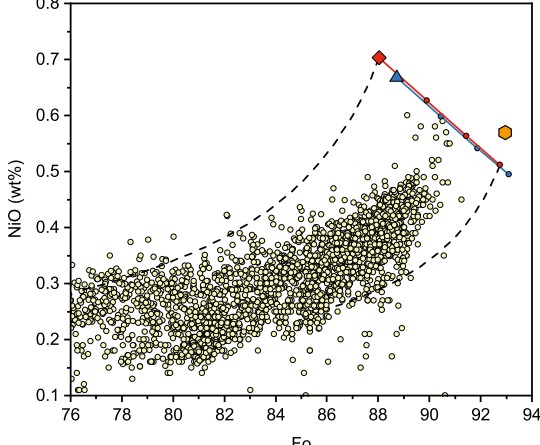

**Fig. 2 | Nickel contents of the modeled Hawaiian parental melts and olivines.**
**a** Modeled Ni contents of the peridotite-derived primary melt and melts derived from it through clinopyroxene and garnet precipitation and subsequent ortho-pyroxene assimilation (see the caption of Fig. 1). **b** Modeled Ni contents of olivines precipitated from the melts in **a**. Large symbols and solid lines with dots in **b** represent the first crystallized olivines. All dash lines show the batch precipitation process of olivine, calculated using Beattie's model (ref. [64]). Data sources of Hawaiian lavas and olivines are the same as in Fig. 1.

previous studies[24] and our experiments. The Ni partition coefficients for garnet and pyroxene were the same as those used in ref. [5]. For olivine, we used equation (4) from ref. [46]. The Ni contents in the source peridotite and orthopyroxene from lithospheric harzburgite are assumed to be 2000 p.p.m. (refs. [5,7,47]) and 722 p.p.m. (Supplementary Fig. 5), respectively. The calculation results were given in Supplementary Tables 2 and 3. Indeed, the partial melt of peridotite has a significantly lower Ni content at any given MgO content than Hawaiian lavas, which was taken as strong evidence for the existence of recycled oceanic crust in the source region[5]. However, our model suggests that such a correlation between peridotite-derived melt and Hawaiian lavas in the Ni-MgO diagram also might be attributed to the decrease in MgO content of the primary magmas in the deep chamber. Precipitation of clinopyroxene and garnet from the primary magmas and the subsequent orthopyroxene assimilation will change the MgO contents of the derivative magmas, keeping their Ni contents approximately constant. Our calculation results show that the parental magmas of Hawaiian lavas, such as the AM-40, the AM-28, and the melts formed from them through orthopyroxene assimilation, should have Ni contents of ~700–730 p.p.m. and MgO contents between 14 and 20 wt%, matching with the Hawaiian whole-rock trend. Figure 2b shows the relationship of Ni and Fo in olivine crystallized from the parental magmas of Hawaiian lavas. Here, it was assumed that the olivine phenocrysts in Hawaiian lavas mainly crystallized at low pressures. Therefore, the calculation was conducted by using the 1 bar experimental Ni partition coefficients between olivine and melt[48]. The high Ni contents in the first-crystallized olivine from the parental magmas are consistent with those of most Ni-rich olivine phenocrysts from Hawaiin lavas.

### Rare earth element concentrations in Hawaiian magma

The rare earth element (REE) patterns of Hawaiian lavas have been calculated according to our model by assuming that the composition of the source peridotite is similar to a 75:25 mixture of primitive mantle[14] and depleted mantle[49]. High-pressure (3.0 GPa) partition coefficients of REE between minerals and melt from peridotite[49] and Fe-rich picrite[50] were used for the calculations of REE concentrations in the melt of source peridotite and the fractionation melt of P19, respectively. The modeling results are presented in Fig. 3 and Supplementary Table 4. The calculation results of the AM-40/AM-28 and the melt derived from them by orthopyroxene assimilation are consistent with the compositions of Hawaiian lavas within a 20% margin of error. However, it is worth noting that the consistency is better for the

light REEs compared to the heavy REEs (Fig. 3a). The slightly lower concentrations of heavy REEs in our solutions may be attributed to the decomposition of the precipitated garnet, which will release heavy REEs to the magma, while keeping the light REEs almost unchanged. Although garnet is stable in the picritic melt at pressures above 2 GPa (ref. [51]), it will react with surrounding olivine, when they are in contact with each other, to form spinel, orthopyroxene, and clinopyroxene[52]. The rising magma may also carry a small amount of garnet, which will decompose at lower pressures. Figure 3b shows the model solutions by assuming that 10 wt% of precipitated garnet is decomposed.

### Genesis of Hawaiian alkaline lavas

As shown in Supplementary Fig. 1, the melts from the second step of our model have a similar composition to the post-shield alkaline lavas. Although the recycled oceanic crust was thought to be necessary for the explanation of the isotopic characteristics of Hawaiian alkaline lavas (ref. [53] and references therein), its major and trace elements data can be explained without such a contribution. It is also consistent with the recent experimental study on the concentrations of Mn and Ni in the early-crystallized olivines from basaltic melt[54].

Silica-poor garnet pyroxenites with compositions like those of peridotite-derived melts are potential source materials of Hawaiian lavas. The low-degree partial melt of this kind of pyroxenite could also produce Hawaiian lavas through orthopyroxene assimilation according to our experimental results and previous studies[22,51]. However, pure pyroxenite is too dense to form a plume. Density calculation results show the pure lherzolitic plume, with an excess temperature of 200 °C and about 8 wt% of pyroxenite in it, has a neutral buoyancy to normal mantle[55]. Considering that the fast rising plume needs a relatively large positive buoyant, the proportion of pyroxenite in such a plume should be much smaller than 8 wt%. In addition, the melt fraction of lherzolite in such a plume should be ~20 wt% to 30 wt% (refs. [24,56]). In this case, the primary magma of Hawaiian lava, which is mainly composed of peridotite-derived melt, also needs to experience a high degree of fractionation.

The precipitation of clinopyroxene and garnet mainly occurs in the deep chamber (Fig. 4). Small amounts of them may be entrained by the rising magma. The clinopyroxene will be out of equilibrium and react with the highly silica-undersaturated host magmas, while the garnet will decompose at low pressures. Thus, none of them appears in the erupted Hawaiian lavas[11]. At shallower depths, melt and harzburgite reactions enhance the migration rate of magmas and result in the entrainment of large amounts of xenocrysts. Because olivine is the only

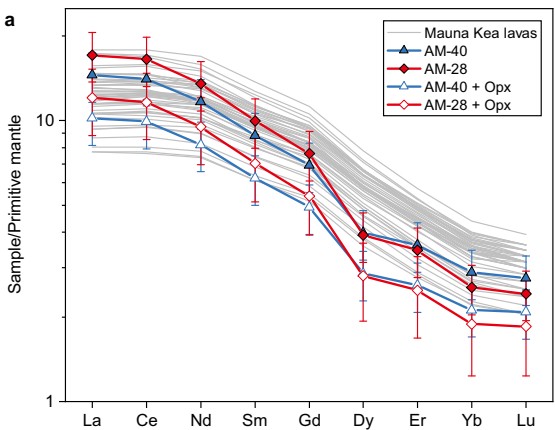

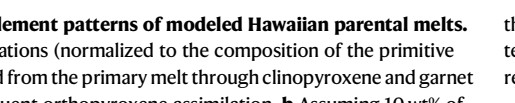

**Fig. 3 | The rare earth element patterns of modeled Hawaiian parental melts.**
**a** Modeled REE concentrations (normalized to the composition of the primitive mantle[14]) of melts derived from the primary melt through clinopyroxene and garnet precipitation and subsequent orthopyroxene assimilation. **b** Assuming 10 wt% of the precipitated garnet has decomposed and released HRREs to the magma (see text). Symbols are drawn with 20% error bars. Data of the Mauna Kea lavas are from ref. [10] with MgO > 14 wt%.

mineral that is saturated within the magmas at shallow depths of less than 45 km (refs. [1,3]), the xenocrysts in Hawaiian lavas are predominantly composed of olivine.

## Methods

### Starting material

A synthetic starting glass material, labeled P19, was used in our experimental studies. As shown in Supplementary Table 1, P19 is similar in composition to the partial melt of mantle peridotite at 3.0 GPa (Run #30.14 of ref. [24]). P19 was prepared by mixing appropriate amounts of pulverous reagents ($SiO_2$, $TiO_2$, MgO, $Fe_2O_3$, $Al_2O_3$, MnO, $P_2O_5$, $Cr_2O_3$, $CaCO_3$, $Na_2CO_3$, $K_2CO_3$) in natural basalt BCR-2. Oxides and carbonate reagents were preheated at 1000 °C and 400 °C, respectively, to remove adsorbed water before weighing. To facilitate equilibration between minerals and melt, the P19 mixture was heated at 1650 °C in the air for 10 min and afterward quenched to glass by submerging it into water. The glass was ground in agate mortars for several hours to a grain size of <5 µm. Then, the glass powders were heated at 900 °C in a graphite-carbon oxide (CCO) buffer for 6 h. The final powders were ground in an agate mortar for several hours with ethanol and dried for >24 h in a 200 °C oven. FTIR analyses show that the P19 glass contains ~0.33 wt% $H_2O$ (Supplementary Fig. 6), which is similar to that in the primary Hawaiian magmas[13,57,58]. The water is likely to be absorbed by the glass during quenching in water.

### Experimental methods

The experiments were conducted in an 1000-ton Walker-type multi-anvil press by using an 18/12 cell assembly (Supplementary Fig. 2). Numerals before and after the slashes are edge sizes of an octahedral pressure medium and truncated corner of a tungsten carbide cube, respectively. The starting material was encapsulated within graphite capsules (inner diameter 1.5 mm, height 2.0 mm), which were subsequently placed in Pt capsules (outer diameter 3.0 mm, inner diameter 2.6 mm, height 3.0 mm). A layer of glassy carbon spheres was loaded at the top of the open capsule to extract melt for precise composition measurement[59–61]. Before final sealing by laser welding machine, the capsules were heated to 200 °C in a vacuum oven for 12 h to keep samples from absorbing water. Type C ($W_5Re$-$W_{26}Re$) thermocouples were used to measure the run temperatures. For these experiments, temperature and pressure gradients across the specimen size are expected to be <25 °C and 0.1 GPa, respectively, based on previous calibration experiments in our laboratory. Experiments were terminated by turning off the power to the heating furnace. The recovered specimen was cut parallel to the

specimen cylindrical axis slightly off the center using a low-speed diamond saw. The large part was mounted in epoxy and polished to expose the center of the sample. Capsules containing carbon spheres were reimpregnated with epoxy once exposed to prevent plugging out from the capsule during polishing.

### Analytical methods

Microstructures of the recovered samples were examined using a FEI Quanta 400 scanning electron microscope at China University of Geosciences. Quantitative analyses of major element compositions of minerals and melts were conducted using a JEOL JXA-8100 electron microprobe at the State Key Laboratory of Submarine Geosciences, State Oceanic Administration, China. Data were corrected using a modified ZAF (atomic number, absorption, fluorescence) correction procedure. The following standards were used for quantification: Jadeite (Na), Rutile (Ti), Pyrope Garnet (Al, Fe), Rhodonite (Mn), Diopside (Si, Mg, Ca), Eskolaite (Cr), Sanidine (K) and Apatite (P). The analyses were conducted with an accelerating voltage of 15 kV, a beam size of 1–5 µm for minerals and 10–20 µm for quenched melts, and a beam current of 20 nA for minerals and 10 nA for melt, respectively. Measured elements were counted for 10 s on peaks and 5 s on the background on each side of the peak. In the low-temperature experiments, quenched glass in the pores of carbon spheres was targeted for analysis. The overlap of the beam with carbon spheres resulted in lower major element totals (>70%) with consistent element ratios. Hence, the melt compositions from all experiments were normalized to 100% (refs. [60–62]) for comparison with natural samples. The FTIR spectra of P19 glass were measured in the wavenumber range of 4000–5500 cm$^{-1}$, using a Nicolet 6700 spectrometer at China University of Geosciences. The P19 glass was first doubly polished to a thickness of 152 µm, then dried in a 200 °C vacuum oven for >24 h to remove surface water. Each FTIR spectrum was obtained with a $100 \times 100$ µm$^2$ aperture, an accumulation of 256 scans, and a resolution of 4 cm$^{-1}$. The background data was collected right after each measurement of the sample.

### Acquisition of chemical equilibrium

To facilitate the equilibrium of the run products, a homogeneous glass was used as starting material. Three pieces of evidence indicate that the run products have attained equilibrium. First, no systematic variations in mineral composition were found across the width or length of the experimental charges. Second, the residual sums of squares in the mass balance calculations are small (from 0.02 to 0.40). Third, the

**a**

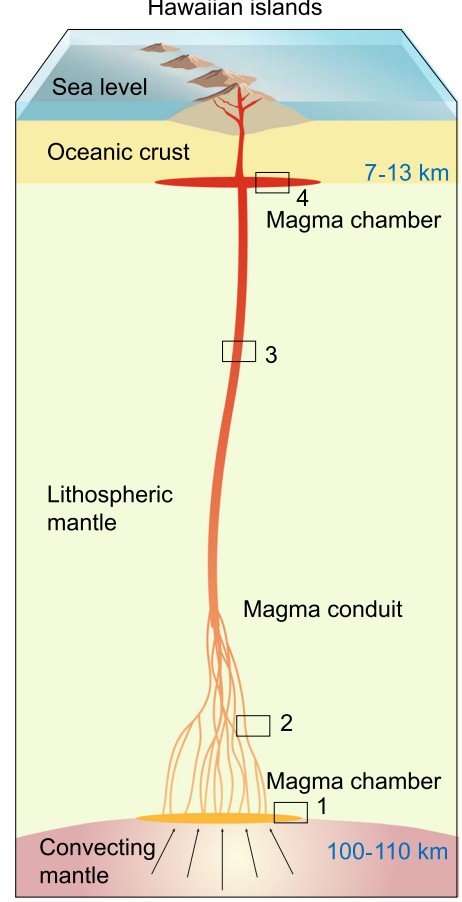

**b**

Fig. 4 | **Schematic diagram of the model for the formation of Hawaiian shield basalts. a** schematic illustration of the high-pressure crystallization model. **b** details of this model. 1, deep magma chamber; 2, deep magma channels; 3, shallower magma channels; 4, crustal magma chamber. In the deep magma chamber and channels (1 and 2), magma precipitates large amounts of clinopyroxene and garnet due to temperature decrease. A thin layer composed of olivine, clinopyroxene, and garnet is formed between the magma and the surrounding lithospheric harzburgite, isolating them from further reaction. In the shallower magma channels and crustal chamber (3 and 4), magma assimilates orthopyroxene from the harzburgite, while olivine becomes the sole mineral saturated with the magma.

Fe-Mg exchange coefficient between olivine and melt ranges from 0.31 to 0.41, which is consistent with those of equilibrated olivine-melt pairs[63].

### Evaluation of the Fe-loss

The Pt-Graphite double capsule can prevent Fe-loss during experiments efficiently because the sample does not directly in contact with the Pt capsule. The small residuals in the mass balance calculations and the reasonable olivine-melt Fe-Mg exchange coefficient values also indicate negligible Fe-loss.

## Data availability

All data generated or analyzed in this study are provided in the Supplementary Information. The data for comparison in this study are available from the GEOROC database (http://georoc.mpch-mainz.gwdg.de/georoc/).

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

## Acknowledgements

We thank Giulio Borghini for the comments on the manuscript. We also thank Dr. Yanfei Zhang for experimental technical assistance; Dr. Wei Li and Dr. Haijun Xu for help in scanning electron microscope analyses; and Dr. Jihao Zhu for technical support in electron probe microanalyzer analyses. This work was supported by funds from the National Key Basic Research Program of China (No. 2015CB856101 to Z.J.), the National Key Research and Development Program of China (No. 2016YFC0600204 to C.W.), the National Natural Science Foundation of China (No. 41530211 and 41872061 to C.W.) and the MOST Special Fund from the State Key Laboratory of GPMR (MSFGPMR02-2 to Z.J.).

## Author contributions

C.W. and J.Y. designed the project. J.Y. conducted the experiments and wrote the manuscript. All authors contributed to the discussion of the results and revising of the manuscript.

## Competing interests

The authors declare no competing interests.
