## [Peer Review File · Nature Communications]

Genesis of Hawaiian lavas by crystallization of picritic magma in the deep mantleREVIEWER COMMENTS

Reviewer #1 (Remarks to the Author):

Dear Editor,

I went through the manuscript NCOMMS-21-00964-T:

"Genesis of Hawaiian lavas by crystallization of picritic magma in the deep mantle" by Junlong Yang, Chao Wang and Zhenmin Jin.

In this paper, Authors propose a petrogenesis model, partially based on new experimental data, to explain the origin of Hawaiian alkali and tholeiitic basalts starting from a picritic melt produced by peridotite partial melting through variable degrees of high-pressure fractional crystallization followed by opx assimilation due to reactive percolation through harzburgite at moderate depths. In order to simulate the first step of the model (high-P segregation in deep magma chamber), they have experimentally provided the melting phase relations and the major elements melt composition derived at 3 GPa from a selected peridotite-derived melt composition. The step of orthopyroxene assimilation via melt-harzburgite reaction is instead calculated. The article is logically structured, well written and concise enough for the format of the journal. Experimental techniques and data are not innovative but well consistent.

This work represents a very nice contribution to the origin of Hawaiian lavas and it is of course of large interest for the scientific community. I think it could be taken into account for the publication on Nature Comm. after a bit of revision mostly addressing few main points and other minor changes.

Although the proposed model is qualitatively convincing it has some critical aspect from the a more quantitative, and dynamically, point of view:

1) The model assumes high extents of garnet+cpx crystallization occurring in deep magma chamber. However, as stated by Authors the very high flux of Hawaiian magmas (more than $8 \text{ m}^3 \text{ s}^{-1}$) requires an abnormally high-degree partial melt of peridotite that is not in agreement with the assumption that Hawaiian primary magmas were formed by low degrees of melting of fertile garnet

herzolite; the weakness here is that on one hand the model requires high melt volume from peridotite partial melting, and on the other hand the high-pressure segregation would form a very huge amount of cpx+garnet rocks that could consist of a permeability barrier. In a more dynamic model, rising peridotite-derived melts should react with this thick mafic layer. Authors can provide an explanation to consider this aspect of the model?

2) After the high-pressure fractionation, the rising melts react with harzburgite assimilating orthopyroxene (and crystallizing olivine? or cpx?). This process occurs in response to the migration of these melts through the lithospheric mantle entirely formed by harzburgite. The model does not specify in which extent the transient melts assimilate opx and if they precipitate other phases, such as olivine or cpx. However, evolved melts generated at rather low temperature (down to 1200°C), are mostly expected to crystallize by reaction with depleted peridotite (please see Wang et al., 2019, and references therein), leaving open the question: is the volume of melt increased or decreased after reaction with harzburgite? From a dynamic point of view, how the harzburgite-bearing lithospheric mantle is modified by the melt migration? Is it still a harzburgite after continuous melt migration?

3) Authors begin the introduction saying that a valid model must match the major and trace element characteristics of Hawaiian lavas. Can the proposed model account for the trace element evolution of the melts starting from the primary peridotite-derived liquid? Which are the expected trace element modifications of the melts according with the high-P fractional crystallization and the subsequent interaction with the harzburgites?

Other minor comments are contained in the pdf file of the manuscript I attached to this review.

Reviewer #2 (Remarks to the Author):

Dear Editor,

Yang et al. present results from new experiments on high pressure (3 GPa) crystallisation of a primary mantle melt composition. They extract the phase relationships as a function of temperature. By comparing the composition of the melt with the composition of Hawaiian lavas they suggest a new deep-crystallisation model for the formation of primitive Hawaiian basalts. This model (supported by the experiments) can more satisfactorily match the major element systematics of Hawaiian lavas than can pristine mantle melts.

The experimental results are a valuable and interesting contribution to the literature, and the novel model they inspire is also of interest. My expertise is not in experimental petrology, but with that caveat, I detect no flaws in the methods and data presented here. However, given the weight that is placed on the presentation of the model, I think further validation against other geochemical and geophysical observations is required. If a more comprehensive analysis is performed, I think this work could be suitable for publication in Nature Communications, and will have broad general interest. The quality of the figures is good. I found the text very confusing in some places, which I have highlighted below. I would also add that I have limited expertise of Hawaiian volcanism, and so I am not able to fully assess whether the existing Hawaii-specific literature is appropriately credited.

Detailed comments below.

1. Is this model consistent with trace elements? As the manuscript says itself on lines 25-26, "A valid model must match the major and trace element characteristics of all Hawaiian lavas," yet trace elements are not discussed further in the manuscript. Crystallisation of garnet and clinopyroxene are likely to have a significant effect on the trace element concentrations of the melts, and given the simplicity of estimating this effect, I think it necessary to validate the model against the trace element observations.
2. Are there any geophysical observations that have any bearing on the likely presence of melt at the base of the lithosphere? E.g., seismic velocity, or conductivity? Or equally, are there geophysical observations that definitively rule out the presence of such a magma body. As for the inclusion of trace elements, I think consideration of such observations may provide additional validation.
3. Does the presence of water in the experiments make them less applicable to Hawaiian magmas? On lines 74-76 the manuscript describes the presence of water (indicated by the crystallisation of phlogopite). Clearly Hawaiian magmas are not anhydrous, but water can have a significant effect on phase stability. I think it would be helpful here if a quantitative estimate of the water content was provided, and compared with the likely water content of primitive Hawaiian magmas.
4. "Disappearance" vs instability of orthopyroxene. On lines 76-78 the manuscript describes the disappearance of orthopyroxene in lower-temperature experiments, ascribing it to a reaction between it and melt to form clinopyroxene. This implies those particular experiments had orthopyroxene at some point, but surely your only observation is its absence at the end of the experiment? Is this not better described as instability? And if it is unstable, then there is no reaction.
5. A thermal gradient in the deep magma chamber (lines 85-88)? The manuscript compares the thermal state of magmas in the chamber when they are "close to the lithospheric mantle" and those that are "relatively far from the lithospheric mantle". Given the fuzziness of the lithosphere-asthenosphere boundary in practice, does this description have any meaning on the length scales of magma chambers? Even if this did, would convection not act to homogenise the temperatures within the magma? And wouldn't any temperature gradient inherited externally be very small compared with any temperature changes in conductive boundary layers at the edges of the chamber?
6. Using mixtures of melts from one magma chamber (lines 88-93). Wouldn't convection act to homogenise most of the magma held in the chamber? Why take mixed melts, will single melts not fit the data? This is the part of the model I find least plausible, and so I think it would benefit from more explanation and defence.
7. Method description needed for the olivine crystallisation calculations (lines 119-125). How was the liquid line of descent calculated? What value of K_d was used to estimate F_0 ?

8. Vague statement on lines 133-134. I don't understand what you mean by "...saturated with lithospheric harzburgite..." and "...where the thermal divide is effective...". It would be useful for the reader to add more explanation here.
9. Olivine-cpx-garnet layer formation (lines 144-145). The text implies there is certainty in this process derived from the experiments, but I don't understand how the experiments have any bearing on this. However, I can see that such a result is feasible, but the text should be adjusted to reflect this.
10. Magma flux explained by a deep magma chamber (lines 158-161). I don't understand how a deep magma chamber has any bearing on magma flux, let alone being an alternative explanation for it? This paragraph makes no sense to me whatsoever- more explanation is needed.
11. Explanation for the lack of clinopyroxene and garnet macrocrysts in lavas (lines 163-171). The manuscript makes the argument that incorporation of clinopyroxene and garnet macrocrysts is unlikely, and even if they are incorporated, they are likely to be resorbed. However, the manuscript previously argued that the presence of clinopyroxene-garnet xenoliths in erupted material is evidence for the model presented here. This seems contradictory to me.
12. Effect of water and potassium on melting temperature (lines 185-186). The manuscript says they "may" decrease the melting temperature. At the very least, water will DEFINITELY decrease the melting temperature, as is very widely documented.
13. Density argument against substantial amounts of pyroxenite in the plume. This argument is made without citations, yet it is not new. Shorttle et al. (2014, EPSL) make the same argument, for example.
14. EPMA analytical techniques (lines 239-249). How was the probe calibrated? Were secondary standards run? This information would be useful to include.

Reviewer #3 (Remarks to the Author):

This experimental study of the genesis of Hawaiian basalts is long overdue. The previous studies were done more than two decades ago and at pressure that were too low to be appropriate. Thus, this new study is a welcome addition.

The paper is well written for one with only Chinese authors. It is also well organized in its content. They have chosen an important issue to examine. Unfortunately, they have fallen into the trap of thinking picrites are representative of primary melts and have relied on the GEOROC database for their comparison. It is well known that that database has lots of errors and it mixes data from many different volcanoes. The authors should focus on one volcano such as Mauna Loa or Kilauea.

Next, I should emphasize this review is focused on petrology of Hawaiian basalts and does not address the experimental methods or data. Hopefully, another reviewer can comment on those parts.

My comments are added to the text in the attached file. The authors are asked to see them there. One of the main concerns is ignoring the diffusion effects on olivine in their modeling. The authors should see the paper by Lynn et al. 2017 (Nickel variability in Hawaiian olivine: Evaluating the relative contributions from mantle and crustal processes. *American Mineralogist*, 102, 507-518, doi.org/10.2138/am-2017-5763). This paper documents the effects of diffusion on olivine composition.

Also, there is NO geophysical evidence for the deep magma chamber that they invoke in their model. No chamber is needed and the reactions can occur during ascent (see Stolper et al., 2004, G3).

The authors get into hot water when they try to apply their model to Hawaiian alkalic rocks, which are isotopically distinct from Hawaiian tholeiites.

Numerous other comments are given on the attached manuscript.

Point-by-point response file

Response to Reviewer #1

- 1) The model assumes high extents of garnet+cpx crystallization occurring in deep magma chamber. However, as stated by Authors the very high flux of Hawaiian magmas (more than $8 \text{ m}^3\text{s}^{-1}$) requires an abnormally high-degree partial melt of peridotite that is not in agreement with the assumption that Hawaiian primary magmas were formed by low degrees of melting of fertile garnet lherzolite; the weakness here is that on one hand the model requires high melt volume from peridotite partial melting, and on the other hand the high-pressure segregation would form a very huge amount of cpx+garnet rocks that could consist of a permeability barrier. In a more dynamic model, rising peridotite-derived melts should react with this thick mafic layer. Authors can provide an explanation to consider this aspect of the model?

Reply: The conclusion “The very high flux of Hawaiian magmas (more than $8 \text{ m}^3\text{s}^{-1}$) requires an abnormally high-degree partial melt of peridotite.” is cited from Sobolev et al. (2005, Nature). However, after re-check the data of Hawaiian swell (e.g. Ribe and Christensen, 1999; Van Ark and Lin, 2004; Vidal and Bonneville, 2004), we find that this conclusion may not be accurate. The plume volume flux (Q_p), the melting degree (F) and the magma flux (Q_v) can be linked as: $Q_v = Q_p \times F$. It is widely accepted that the swell is supported by the hot plume residue, which result in density anomaly in the mantle (Ribe and Christensen, 1994, 1999; Cserepes et al., 2000; Cadio et al., 2012). In this case, the effective buoyancy is mainly caused by thermal expansion of plume residual and Q_v , Q_p and B can be linked as:

$$B = \alpha \times \rho_m \times \Delta T \times (Q_p - Q_v)$$

where $B = 3500 \text{ kg/s}$ is the buoyancy flux (Ribe and Christensen, 1999), $\alpha = 3.5 \times 10^{-5} \text{ }^\circ\text{C}^{-1}$ is the thermal expansion coefficient, $\rho_m = 3300 \text{ kg/m}^3$ is the density of reference mantle, which is assumed to be 100% lherzolite. Assuming that the average excess temperature (ΔT) is 200°C , we obtain from the above equation that $Q_p = 160 \text{ m}^3\text{s}^{-1}$. In this case, the partial melting degree of the source peridotite is $\sim 5 \text{ wt}\%$, which is not so abnormally. After careful consideration, we decided to delete this paragraph in the revised manuscript.

Sobolev, A. V., Hofmann, A. W., Sobolev, S. V. & Nikogosian, I. K. An olivine-free

mantle source of Hawaiian shield basalts. *Nature* 434, 590-597 (2005).

Ribe, N. M. & Christensen, U. R. The dynamical origin of Hawaiian volcanism. *Earth and Planetary Science Letters* 171, 517-531 (1999).

Van Ark, E. & Lin, J. Time variation in igneous volume flux of the Hawaii-Emperor hot spot seamount chain. *Journal of Geophysical Research* 109, B11401 (2004).

Vidal, V. & Bonneville, A. Variations of the Hawaiian hot spot activity revealed by variations in the magma production rate. *Journal of Geophysical Research* 109, B03104 (2004).

Ribe, N. M. & Christensen, U. R. Three-dimensional modeling of plume-lithosphere interaction. *Journal of Geophysical Research* 99, 669-682 (1994).

Cserepes, L., Christensen, U. R. & Ribe, N. M. Geoid height versus topography for a plume model of the Hawaiian swell. *Earth and Planetary Science Letters* 178, 29-38 (2000).

Cadio, C., Ballmer, M. D., Panet, I., Diament, M. & Ribe, N. New constraints on the origin of the Hawaiian swell from wavelet analysis of the geoid to topography ratio. *Earth and Planetary Science Letters* 359-360, 40-54 (2012).

- 2) After the high-pressure fractionation, the rising melts react with harzburgite assimilating orthopyroxene (and crystallizing olivine? or cpx?). This process occurs in response to the migration of these melts through the lithospheric mantle entirely formed by harzburgite. The model does not specify in which extent the transient melts assimilate opx and if they precipitate other phases, such as olivine or cpx. However, evolved melts generated at rather low temperature (down to 1200°C), are mostly expected to crystallize by reaction with depleted peridotite (please see Wang et al., 2019, and references therein), leaving open the question: is the volume of melt increased or decreased after reaction with harzburgite? From a dynamic point of view, how the harzburgite-bearing lithospheric mantle is modified by the melt migration? Is it still a harzburgite after continuous melt migration?

Reply: The reaction between rising melt and mantle harzburgite at shallower depth can be taken as a combination of two processes. One is crystallizing of $Ol \pm Opx \pm Sp \pm Cpx$ from the melt (Wagner and Grove, 1998), which depends on the degree of temperature decreasing of melt during reaction, instead of its absolute temperature. The melt will become “superheated” during adiabatic ascent due to the lowering of its liquidus temperature. In addition, the relative faster migration rate of magmas at

shallower depth will limited the heat exchange between the rising melt and surrounding harzburgite (e.g. Ryan, 1988; Wright and Klein, 2006). So, we believed that crystallizing of $Ol \pm Opx \pm Sp \pm Cpx$ is not a dominant process. The other one is the Opx assimilation and Ol crystallization causing by disequilibrium of chemical composition (Daines and Kohlstedt, 1994). The ratio of orthopyroxene assimilated to olivine crystallized (Ma/Mc) for natural silicate system likely to range from 1.2 to 1.4 (Kelemen, 1990). So, it seems that the volume of melt should be slightly increased after the reaction. Such reaction will result in the formation of dunite magma channels (Lundstrom et al., 2000).

Wagner, T. P. & Grove, T. L. Melt/harzburgite reaction in the petrogenesis of tholeiitic magma from Kilauea volcano, Hawaii. *Contributions to Mineralogy and Petrology* 131, 1-12 (1998).

Ryan, M. P. The mechanics and three-dimensional internal structure of active magmatic systems: Kilauea Volcano, Hawaii. *Journal of Geophysical Research* 93, 4213-4248 (1988).

Wright, T. L. & Klein, F. W. Deep magma transport at Kilauea volcano, Hawaii. *Lithos* 87, 50-79 (2006).

Daines, M. J. & Kohlstedt, D. L. The transition from porous to channelized flow due to melt/rock reaction during melt migration. *Geophysical Research Letters* 21, 145-148 (1994).

Kelemen, P. B. Reaction Between Ultramafic Rock and Fractionating Basaltic Magma I. Phase Relations, the Origin of Calc-alkaline Magma Series, and the Formation of Discordant Dunite. *Journal of Petrology* 31, 51-98 (1990).

Lundstrom, C. C., Gill, J. & Williams, Q. A geochemically consistent hypothesis for MORB generation. *Chemical Geology* 162, 105-126 (2000).

- 3) Authors begin the introduction saying that a valid model must match the major and trace element characteristics of Hawaiian lavas. Can the proposed model to account for the trace element evolution of the melts starting from the primary peridotite-derived liquid? Which are the expected trace element modifications of the melts according with the high-P fractional crystallization and the subsequent interaction with the harzburgites?

Reply: We have calculated the REE concentrations in the magmas according to our model in the revised manuscript (Lines 189-206 and Fig. 3).

- 4) Line 27: In this comparison, Authors missed to include the data of partial melting experiments on silica-deficient pyroxenite (e.g. Hirschmann et al., 2003 *Geology*), which have major element composition significantly different from melt produced by SE pyroxenites and eclogites (please see also Lambart et al., 2013). Is there any specific reason to exclude these data?

Reply: The silica-deficient pyroxenites is similar in composition to those of peridotite-derived melts, so we have discussed its possible contribution to the genesis of Hawaiian lavas at the end of the manuscript (Lines 216-226).

- 5) Line 85: How much is assumed to be the temperature of the hosting lithospheric mantle?

Reply: We assumed that the temperature of the bottom of lithospheric mantle is about 1100°C (e.g. Kawakatsu et al., 2009, *Science*).

Kawakatsu, H. et al. Seismic Evidence for Sharp Lithosphere-Asthenosphere Boundaries of Oceanic Plates. *Science* 324, 499-502 (2009).

- 6) Line 95: What does it mean 0-30% of opx? 0-30% of what? Please specify this amount.

Reply: Have been fixed to 0-30 wt%.

- 7) Line 97: I do not completely agree here; the amount of olivine crystallization controls the volume of melt resulting from the interaction with the harzburgite. I think it would be a critical parameter.

Reply: Thanks for the reviewer's suggestion. This is really not a proper description. So, we have modified this part in the revised manuscript (Lines 141-149).

- 8) Lin 110: This is not clear to me. The primary peridotite-derived melt is expected to have low Ni concentrations, as olivine is still in the residue. I understand that the segregation of Ni-almost-free phases as garnet and cpx will increase the Ni of evolved melt, but what is the role of MgO variation in the melt? Maybe I lost here but the point is not very clear.

Reply: Some Hawaiian lavas are enriched in NiO at given MgO content in comparison with peridotite derived melt (Fig. 2a in Sobolev et al., 2005). However, from the other point of view, these lavas can be taken as deficient in MgO in comparison with peridotite-derived magmas at given NiO. The segregation of Ni-almost-free phases as grt and cpx will slightly increase the NiO and decrease the MgO of evolved melt, which matches the composition of these Hawaiian lavas in the Ni-MgO diagram (Fig. 2 in our manuscript).

Sobolev, A. V., Hofmann, A. W., Sobolev, S. V. & Nikogosian, I. K. An olivine-free mantle source of Hawaiian shield basalts. *Nature* 434, 590-597 (2005).

- 9) Line 125: Here I suggest to compare the results of these calculation with those from Matzen et al. (2017) *Nature* who come to similar conclusions.

Reply: The calculated Ni content in olivine depends on the Ni concentration in primary magma and the Ni partition coefficient between olivine and melt. The Ni partition coefficient used in this study is given by the equation (10) in Wang et al. (2008), which is similar with that given by the equation (4) in Matzen et al. (2017, *CMP*). The conclusion is similar like Matzen et al. (2017, *Nature geoscience*).

Wang, Z. & Gaetani, G. A. Partitioning of Ni between olivine and siliceous eclogite partial melt: experimental constraints on the mantle source of Hawaiian basalts. *Contributions to Mineralogy and Petrology* 156, 661-678 (2008).

Matzen, A. K., Baker, M. B., Beckett, J. R., Wood, B. J. & Stolper, E. M. The effect of liquid composition on the partitioning of Ni between olivine and silicate melt. *Contributions to Mineralogy and Petrology* 172, 3 (2017).

Matzen, A. K., Wood, B. J., Baker, M. B. & Stolper, E. M. The roles of pyroxenite and peridotite in the mantle sources of oceanic basalts. *Nature Geoscience* 10, 530-535 (2017).

- 10) Line 138: Keshav et al. (2007) documented olivine-bearing clinopyroxenites, which is the difference from their results in term of phase relations?

Reply: The primary magma is formed by reactions of “7 Ol + 68 Cpx + 25 Grt = 84 melt + 16 Opx” with melt fraction from 0 to 10% and “6 Ol + 94 Cpx = 64 melt + 36 Opx” at melt fraction from 10-23% at 3 GPa (Table 9 in Walter, 1998). Therefore, the

primary magma should be out of equilibrium with Opx and reacted with the latter to form Ol + Cpx at the initial stage of fractionation. Our experiments were conducted by using primary magma as starting material without additional Opx. So, the precipitation is composed of Grt and Cpx in the ratio about 1:1. However, the primary magma is surrounded by mantle harzburgite in the nature system. So, we believed that the reaction of primary magma with harzburgite in the deep magma chamber should form clinopyroxenite.

Walter, M. J. Melting of Garnet Peridotite and the Origin of Komatiite and Depleted Lithosphere. *Journal of Petrology* 39, 29-60 (1998).

11) Line 142: I do not understand if the model assumes also precipitation of garnet and cpx concomitant to the opx assimilation.

Reply: Please see the reply under the question NO. 10.

12) Line 149: Are the transient melt expected to change their trace element composition via interaction with the harzburgites?

Reply: Yes. We have made such discussion in the REE concentration calculation (Lines 189-206, Fig. 3 and Supplementary Table 4).

13) Line 161: However, the magma chamber needs to be replenished from depth, therefore the model still would require high volume of peridotite-derived melt coming into the chamber.

Reply: Please see the reply under the first question.

14) Line 168: This mostly depend on the ratio between assimilated and crystallized mass; here Authors should explain better which kind of process drives the melt-harzburgite interaction also estimating how the reaction impacts on the volume of melt going out of the system.

Reply: More detailed discussion and explanation are added in the revised manuscript (Lines 126-134).

15) Line 384: Supplementary information on the experimental data should include a brief discussion on the approach to the equilibrium in the experiments.

Reply: The equilibrium of the experiments is discussed in experimental result (Lines 82-85).

Response to Reviewer #2

- 1) Is this model consistent with trace elements? As the manuscript says itself on lines 25-26, “A valid model must match the major and trace element characteristics of all Hawaiian lavas,” yet trace elements are not discussed further in the manuscript. Crystallisation of garnet and clinopyroxene are likely to have a significant effect on the trace element concentrations of the melts, and given the simplicity of estimating this effect, I think it necessary to validate the model against the trace element observations.

Reply: We have calculated the REE concentrations in the magmas according to our model in the revised manuscript (Lines 189-206 and Fig. 3).

- 2) Are there any geophysical observations that have any bearing on the likely presence of melt at the base of the lithosphere? E.g., seismic velocity, or conductivity? Or equally, are there geophysical observations that definitively rule out the presence of such a magma body. As for the inclusion of trace elements, I think consideration of such observations may provide additional validation.

Reply: The near global observation of sharp drop in seismic velocity near the base of the lithosphere has been attributed in some locations to melt accumulation (Havlin et al., 2013, EPSL, and reference therein).

Havlin, C., Parmentier, E. M. & Hirth, G. Dike propagation driven by melt accumulation at the lithosphere–asthenosphere boundary. *Earth and Planetary Science Letters* 376, 20-28 (2013).

- 3) Does the presence of water in the experiments make them less applicable to Hawaiian magmas? On lines 74-76 the manuscript describes the presence of water (indicated by the crystallisation of phlogopite). Clearly Hawaiian magmas are not anhydrous, but water can have a significant effect on phase stability. I think it would be helpful here if a quantitative estimate of the water content was provided, and compared with the likely water content of primitive Hawaiian magmas.

Reply: FTIR analysis shows that the P19 glass contains about 0.33 wt% H₂O (Supplementary Figure 6), which is similar with that in the primary Hawaiian magmas (Clague et al., 1995; Wallace and Anderson, 1998; Stolper et al., 2004). So, we don't

think that the water in the experiments make the results less applicable to Hawaiian magmas.

Clague, D. A., Moore, J. G., Dixon, J. E. & Friesen, W. B. Petrology of Submarine Lavas from Kilauea's Puna Ridge, Hawaii. *Journal of Petrology* 36, 299-349 (1995).

Wallace, P. J. & Anderson Jr, A. T. Effects of eruption and lava drainback on the H₂O contents of basaltic magmas at Kilauea Volcano. *Bulletin of Volcanology* 59, 327-344 (1998).

Stolper, E., Sherman, S., Garcia, M., Baker, M. & Seaman, C. Glass in the submarine section of the HSDP2 drill core, Hilo, Hawaii. *Geochemistry Geophysics Geosystems* 5, Q07G15 (2004).

- 4) “Disappearance” vs instability of orthopyroxene. On lines 76-78 the manuscript describes the disappearance of orthopyroxene in lower-temperature experiments, ascribing it to a reaction between it and melt to form clinopyroxene. This implies those particular experiments had orthopyroxene at some point, but surely your only observation is its absence at the end of the experiment? Is this not better described as instability? And if it is unstable, then there is no reaction.

Reply: Peridotite-derive primary magma is formed by reaction of $\text{Cpx} + \text{Grt} + \text{Ol} = \text{Melt} + \text{Opx}$ (Table 9 in Walter, 1998). Therefore, the fractionated magma should be out of equilibrium with Opx and reacted with the latter to form $\text{Ol} + \text{Cpx} + \text{Grt}$.

Walter, M. J. Melting of Garnet Peridotite and the Origin of Komatiite and Depleted Lithosphere. *Journal of Petrology* 39, 29-60 (1998).

- 5) A thermal gradient in the deep magma chamber (lines 85-88)? The manuscript compares the thermal state of magmas in the chamber when they are “close to the lithospheric mantle” and those that are “relatively far from the lithospheric mantle”. Given the fuzziness of the lithosphere-asthenosphere boundary in practice, does this description have any meaning on the length scales of magma chambers? Even if this did, would convection not act to homogenise the temperatures within the magma? And wouldn't any temperature gradient inherited externally be very small compared with any temperature changes in conductive boundary layers at the edges of the chamber?

Reply: Considering the large temperature difference between the primary magmas and lithospheric mantle, it is reasonable to assume that heat exchange will result in thermal gradient in magma chamber. Then, the density difference causing by temperature difference will generate convection in magma chamber, which in turn result in the homogenization of the composition and temperature of magmas that have experienced different degrees of fractionation.

- 6) Using mixtures of melts from one magma chamber (lines 88-93). Wouldn't convection act to homogenise most of the magma held in the chamber? Why take mixed melts, will single melts not fit the data? This is the part of the model I find least plausible, and so I think it would benefit from more explanation and defence.

Reply: The using of mixture is just base on the consideration of the homogeneization of convection. Because, the homogenizing process is exactly a process that mixing magma from different degrees of fractionation. Actually, many researchers are using the mixture of peridotite partial melts as the primary magma (e.g. sobolev et al., 2007).

Sobolev, A. et al. The Amount of Recycled Crust in Sources of Mantle-Derived Melts. *Science* 316, 412-417 (2007).

- 7) Method description needed for the olivine crystallisation calculations (lines 119-125). How was the liquid line of descent calculated? What value of K_d was used to estimate F_o ?

Reply: The olivine crystallization calculations were carried out using Beattie's model (Beattie et al., 1991). The value of K_d from Beattie et al. (1991) was used to estimate the F_o . This information was mentioned in the caption of Fig. 2.

Beattie, P., Ford, C. & Russell, D. Partition coefficients for olivine-melt and orthopyroxene-melt systems. *Contributions to Mineralogy and Petrology* 109, 212-224 (1991).

- 8) Vague statement on lines 133-134. I don't understand what you mean by "...saturated with lithospheric harzburgite..." and "...where the thermal divide is effective...". It would be useful for the reader to add more explanation here.

Reply: This part has been revised (Lines 103 to 115).

- 9) Olivine-cpx-garnet layer formation (lines 144-145). The text implies there is certainty in this process derived from the experiments, but I don't understand how the experiments have any bearing on this. However, I can see that such a result is feasible, but the text should be adjusted to reflect this.

Reply: The text has been adjusted (Lines 103-115).

- 10) Magma flux explained by a deep magma chamber (lines 158-161). I don't understand how a deep magma chamber has any bearing on magma flux, let alone being an alternative explanation for it? This paragraph makes no sense to me whatsoever- more explanation is needed.

Reply: Many thanks for the suggestion. This paragraph has been deleted in the revised manuscript.

- 11) Explanation for the lack of clinopyroxene and garnet macrocrysts in lavas (lines 163-171). The manuscript makes the argument that incorporation of clinopyroxene and garnet macrocrysts is unlikely, and even if they are incorporated, they are likely to be resorbed. However, the manuscript previously argued that the presence of clinopyroxene-garnet xenoliths in erupted material is evidence for the model presented here. This seems contradictory to me.

Reply: The preserve of clinopyroxene-garnet xenoliths depends on its size and reaction rate with the magma. If the clinopyroxene and garnet are carried by the magmas as discrete grains, they will be resorbed rapidly by the magma at lower pressure. However, if they were originally carried as large xenoliths, they may preserve as a relatively small xenoliths in the erupt lavas.

- 12) Effect of water and potassium on melting temperature (lines 185-186). The manuscript says they "may" decrease the melting temperature. At the very least, water will DEFINITELY decrease the melting temperature, as is very widely documented.

Reply: Thanks for the reviewer's suggestion. This is really not a proper description. This paragraph has been deleted in the revised manuscript.

13) Density argument against substantial amounts of pyroxenite in the plume. This argument is made without citations, yet it is not new. Shorttle et al. (2014, EPSL) make the same argument, for example.

Reply: We have cited the previous studies (Lines 220-222).

14) EPMA analytical techniques (lines 239-249). How was the probe calibrated? Were secondary standards run? This information would be useful to include.

Reply: This information has been included in “Methods” of the revised manuscript (Lines 278-285).

Response to Reviewer #3

- 1) One of the main concerns is ignoring the diffusion effects on olivine in their modeling. The authors should see the paper by Lynn et al. 2017 (Nickel variability in Hawaiian olivine: Evaluating the relative contributions from mantle and crustal processes. *American Mineralogist*, 102, 507–518, doi.org/10.2138/am-2017-5763). This paper documents the effects of diffusion on olivine composition.

Reply: Lynn's diffusion model can explain the high variation of Ni concentration in Hawaiian olivine at given Fo value. However, magma formed by mixing of peridotite-derive melt and eclogite-derived melt will have lower whole rock Ni concentration in comparison with that in most Hawaiian lavas (dash line with black dots in Fig. 2a from Sovbolev et al., 2005).

Sobolev, A. V., Hofmann, A. W., Sobolev, S. V. & Nikogosian, I. K. An olivine-free mantle source of Hawaiian shield basalts. *Nature* 434, 590-597 (2005).

- 2) Also, there is NO geophysical evidence for the deep magma chamber that they invoke in their model. No chamber is needed and the reactions can occur during ascent (see Stolper et al., 2004, G3).

Reply: We have discussed the possibility existence of deep magma chamber in the revised manuscript. Pleases see the discussion in the manuscript (Lines 88 to 97).

- 3) The authors get into hot water when they try to apply their model to Hawaiian alkalic rocks, which are isotopically distinct from Hawaiian tholeiites.

Reply: As shown in Supplementary Figure 1, the melts from the second step of our model have similar composition with that of the post-shield alkaline lavas. Thus, although recycled oceanic crust is thought to be necessary for the explanation the isotopic character of Hawaiian alkaline lavas (Huang et al., 2013, and reference therein), its major and trace elements data can be explained without such a contribution. This is consistent with the recent experimental study on the concentrations of Mn and Ni in the early-crystallizing olivines from basaltic melt (Matzen et al., 2017, *Nature geoscience*).

Huang, S., Blichert-Toft, J., Fodor, R. V., Bauer, G. R. & Bizimis, M. Sr, Nd, Hf and Pb isotope systematics of postshield-stage lavas at Kahoolawe, Hawaii. *Chemical*

Geology 360-361, 159-172 (2013).

Matzen, A. K., Wood, B. J., Baker, M. B. & Stolper, E. M. The roles of pyroxenite and peridotite in the mantle sources of oceanic basalts. *Nature Geoscience* 10, 530-535 (2017).

- 4) Line 24: The petrogenesis of Hawaiian lavas is still hotly debated. Cited papers 8-18 years old.

Reply: This sentence rewrote as: “The petrogenesis of Hawaiian lavas is still not clear.”

- 5) Line 27: It is a bad idea to use the error prone source for a rigorous evaluation. Original references should be used.

Reply: The amounts of original references were too much. So, the data of the Mauna Kea lavas of Hawaii from Hawaii Scientific Drilling Project (Rhodes et al., 2004) has been used to compare with the data from the GEOROC database in revised manuscript Supplementary Figure 1. There is no systematic difference between them.

Rhodes, J. M. & Vollinger, M. J. Composition of basaltic lavas sampled by phase-2 of the Hawaii Scientific Drilling Project: Geochemical stratigraphy and magma types. *Geochemistry Geophysics Geosystems* 5, Q03G13 (2004).

- 6) Line 29: The pressure range of previous experimental data that used in this study.

Reply: The sentence has been changed to ‘..., we use the experimental data with pressures from 3.0 to 4.0 GPa.’ (Lines 31-32).

- 7) Line 34: The range of the low degrees of melting?

Reply: About 5 wt% given by Prytulak and Elliott (2007).

Prytulak, J. & Elliott, T. TiO₂ enrichment in ocean island basalts. *Earth and Planetary Science Letters* 263, 388-403 (2007).

- 8) Line34: ‘fertile garnet lherzolite’ is wrong because most could argue the source is at least somewhat depleted based on Pb, Sr and Nd isotope.

Reply: The ‘fertile’ is deleted.

9) Line 39: Stolper et al. (2004) made this argument based on lava compositions.

Reply: Stolper et al. (2004) has been cited in the revise manuscript (Line 41).

10) Line 49: Not true, Stolper et al., 2004; Eggin et al., 1997.

Reply: This sentence has been revised as “Many previous studies are based on the assumption that the composition of the primary magmas of Hawaiian lavas did not change during transit from the mantle to the crust. However, this may not be accurate (Eggin et al., 1992; Wagner and Grove, 1998; Stolper et al., 2004).” (Lines 51-53).

Eggin, S. M. Petrogenesis of Hawaiian tholeiites: 1, phase equilibria constraints. *Contributions to Mineralogy and Petrology* 110, 387-397 (1992).

Wagner, T. P. & Grove, T. L. Melt/harzburgite reaction in the petrogenesis of tholeiitic magma from Kilauea volcano, Hawaii. *Contributions to Mineralogy and Petrology* 131, 1-12 (1998).

Stolper, E., Sherman, S., Garcia, M., Baker, M. & Seaman, C. Glass in the submarine section of the HSDP2 drill core, Hilo, Hawaii. *Geochemistry Geophysics Geosystems* 5, Q07G15 (2004).

11) Line 54: Not derived from tholeiites. They are related to rejuvenation lavas. See Guest et al., 2020 G3.

Reply: The pyroxenite xenoliths are carried by rejuvenation stage lavas, which doesn't mean that it must be originated from it.

12) Line 71: “~3 wt%”. How was this determined?

Reply: This proportion of olivine in the run products is calculated by mass balance.

13) Line 71: “stable”. What does this mean?

Reply: It means that the proportions of olivine almost do not change with decreasing temperatures.

14) Line 72: When does it appear?

Reply: We can not confirm the precise temperature for the appearance of orthopyroxene from our present experimental results.

15) Line 74: Since mica normally has 4 wt% H₂O, the capsule had lots of water.

Reply: FTIR analysis shows that the P19 glass contains about 0.33 wt% H₂O (Supplementary Figure 6), which is similar with that in the primary Hawaiian magmas (Clague et al., 1995; Wallace and Anderson, 1998; Stolper et al., 2004).

Clague, D. A., Moore, J. G., Dixon, J. E. & Friesen, W. B. Petrology of Submarine Lavas from Kilauea's Puna Ridge, Hawaii. *Journal of Petrology* 36, 299-349 (1995).

Wallace, P. J. & Anderson Jr, A. T. Effects of eruption and lava drainback on the H₂O contents of basaltic magmas at Kilauea Volcano. *Bulletin of Volcanology* 59, 327-344 (1998).

Stolper, E., Sherman, S., Garcia, M., Baker, M. & Seaman, C. Glass in the submarine section of the HSDP2 drill core, Hilo, Hawaii. *Geochemistry Geophysics Geosystems* 5, Q07G15 (2004).

16) Line 85: ?km.

Reply: We think that the deep magma chamber should be located at the depth between 60 km to 90 km.

17) Line 100: This is discussion and does not belong in the results.

Reply: The paragraph of “Ni in Hawaiian melts and olivine” and “High-pressure crystallization model” are both moved to Discussion in the revised manuscript (Lines 151-187).

18) Line 110: What is the Ni content of the minerals in your experiments?

Reply: Our P19 glass is almost Ni-free. Therefore, we can't analysis the Ni concentration in the minerals of our run products by using EPMA.

19) Line 114: Would also change SiO₂.

Reply: We have deleted the word “only” in the revised manuscript (Line 176).

20) Line 115: Evidence for this claim?

Reply: This claim is based on our calculation result (Fig. 3 and Supplementary Table 2 and 3).

21) Line 116-118: Not well known.

Reply: It is the average number of calculated Ni concentrations in each experimental run. The calculated method and result are shown in Lines 162-165 and Supplementary Table 2.

22) Line 124: Evidence?

Reply: This description is based on our calculation result (Fig. 2b and Supplementary Table 3).

23) Line 140: Evidence?

Reply: We have deleted this part in the revised manuscript.

24) Line 140: Magma from even a single eruption can be just variable in composition (Greene et al., 2013, G3).

Reply: We have deleted this part in the revised manuscript.

25) Line 154: Why are they isotopically different (Hinano et al., 2010)?

Reply: Our proposal was made base on the major element composition. It is inconsistent with the isotopic characteristic of Hawaiian alkaline magmas. Since there is no isotope data in our experimental result. We decided to delete this paragraph.

26) Line 163: Evidence for this claim?

Reply: We believed that the primary magmas will first be accumulated in the deep magma chamber. Therefore, its migration rate should be very low in deep lithosphere.

27) Line 168: Evidence for the migration rate of magmas at shallower depths.

Reply: Please see the discussion in the revised manuscript (Lines 126 to 134).

28) Line 170: Reference?

Reply: Reference have been cited in the revised manuscript (Line 236).

29) Line: 173: Why? Not clear how your experiments say this.

Reply: We have deleted this paragraph.

30) Line 180 to 185: This is basic, so why did your experiments do this?

Reply: Although it is not our intention, the occurrence of water in our starting material doesn't make our experimental results less applicable to Hawaiian magmas. After all, Hawaiian magmas are not anhydrous. FTIR analysis shows that the P19 glass contains about 0.33 wt% H₂O (Supplementary Figure 6), which is similar with that in the primary Hawaiian magmas (Clague et al., 1995; Wallace and Anderson, 1998; Stolper et al., 2004).

Clague, D. A., Moore, J. G., Dixon, J. E. & Friesen, W. B. Petrology of Submarine Lavas from Kilauea's Puna Ridge, Hawaii. *Journal of Petrology* 36, 299-349 (1995).

Wallace, P. J. & Anderson Jr, A. T. Effects of eruption and lava drainback on the H₂O contents of basaltic magmas at Kilauea Volcano. *Bulletin of Volcanology* 59, 327-344 (1998).

Stolper, E., Sherman, S., Garcia, M., Baker, M. & Seaman, C. Glass in the submarine section of the HSDP2 drill core, Hilo, Hawaii. *Geochemistry Geophysics Geosystems* 5, Q07G15 (2004).

31) Line 197-208: Totally speculative. Delete.

Reply: We accepted the suggestion of the reviewer and deleted the whole paragraph in the revised manuscript.

32) Line 215: Why BCR-2?

Reply: Preparation of starting materials by using nature sample (BCR-2) and oxides could merge the trace element in to it. Then, the partial coefficient of trace element between minerals and melt can be obtained by using ion Microprobe Analysis in case of necessary.

33) Line 219: How was this confirmed?

Reply: According to our previous unpublished experimental studies, the Fe³⁺/Fe²⁺ ratio in basaltic glass will decreased to lower than 0.1 after being sintered in CCO buffer for

5 hours at 900°C. On the other hand, our high pressure experimental was also conducted in CCO buffer. Here, the “deoxidize iron” is truly not a proper expression. So, we have deleted it in the revised manuscript.

34) Line 233: Reference for this claim?

Reply: This is based on the previous unpublished calibration experiment.

35) Line 233: How quickly quenched?

Reply: The temperature of sample decreased to below 500°C in about 2 seconds.

36) Line 246: Peak? Backgrounds? Very short; this causes large errors. Quick and “dirty” results with low beam current and V short counting times.

Reply: Measured elements were counted for 10 s on peaks and 5 s on the background on each side of the peak (Lines 284-285).

37) Line 249: Poor practice. Report results and then normalize to 100%.

Reply: For individual experiments, analyses of melt overlap carbon sphere with major-element totals less than 100% produced consistent element ratios and hence similar compositions when normalized to 100% (Spandler et al., 2008). For comparative purpose and calculation of average melt composition, melt compositions from all experiments were normalized to 100%, like conducted by Dasgupta et al. (2007).

Spandler, C., Yaxley, G., Green, D. H. & Rosenthal, A. Phase Relations and Melting of Anhydrous K-bearing Eclogite from 1200 to 1600°C and 3 to 5 GPa. *Journal of Petrology* 49, 771-795 (2008).

Dasgupta, R., Hirschmann, M. M. & Smith, N. D. Partial Melting Experiments of Peridotite + CO₂ at 3 GPa and Genesis of Alkalic Ocean Island Basalts. *Journal of Petrology* 48, 2093-2124 (2007).

38) Line 416: Periolation VS open channel?

Reply: We believed that both percolation and small channels should exist in the deep lithosphere.

39) Line 421: Evidence for assimilation.

Reply: Experimental studies provide that the silica-undersaturated magma will assimilate orthopyroxene and crystalize olivine at lower pressures (Daines and Kohlstedt, 1994; Wagner and Grove, 1998).

Daines, M. J. & Kohlstedt, D. L. The transition from porous to channelized flow due to melt/rock reaction during melt migration. *Geophysical Research Letters* 21, 145-148 (1994).

Wagner, T. P. & Grove, T. L. Melt/harzburgite reaction in the petrogenesis of tholeiitic magma from Kilauea volcano, Hawaii. *Contributions to Mineralogy and Petrology* 131, 1-12 (1998).

REVIEWER COMMENTS

Reviewer #1 (Remarks to the Author):

Giulio Borghini
Dipartimento di Scienze della Terra – University of Milano
Italy

Second Review of Manuscript NCOMMS-21-00964-T:

“Genesis of Hawaiian lavas by crystallization of picritic magma in the deep mantle” by Junlong Yang, Chao Wang and Zhenmin Jin

Dear Editor, I went through the manuscript I revised some months ago for Nature. I carefully read the point-by-point responses and all the critical points have been correctly addressed. I like this paper and I'd like to see it published. However, as it wants to be a Nature Communications paper, I think it could be further improved by addressing few other weaknesses I reported in the annotated pdf file here attached. Most of them are request of clarification or suggestions. A major point remains the physical aspect of the model: how do you conceal melt migration within conduits and/fracture (it needs to be specify) along 15 kbar (from 3 GPa, garnet+cpx segregation, to about 1.4 GPa), without any melt modification, and the melt migration via reactive porous flow (if I understand correctly) causing opx dissolution in harzburgite, at shallower (maybe colder?) mantle levels (above 1.4 GPa)?

I'm convinced that after a minor revision this paper will be ready to be published on Nature Communications.

Kind regards,

Giulio Borghini

Reviewer #2 (Remarks to the Author):

In this revised manuscript Yang et al. present further justification for their model by incorporating predictions of the trace element concentrations in the lavas that are generated by their model. As I said in my previous review, I think the work is worthy of publication and is an important contribution to the literature. However, I have some doubts over the new aspect of the model (point 3 below), and some of the discussion is confusing (points 1 and 2).

1. Line 74-75. Reiterating from my previous review that saying “orthopyroxene disappeared” when describing the experiments implies that there was orthopyroxene in the starting material, or it crystallised and then was resorbed. The last sentence in the paragraph says the starting material was “homogeneous hydrous glass”, so it cannot be the first, and since the phase proportions were determined following quenching, it cannot be the second. The response to this comment in my first review (Reviewer 2, point 4 in the rebuttal) did not address this.

2. Lines 103-109. The text cites experimental literature that shows in the pressure range under consideration mantle melting proceeds by the reaction olivine + clinopyroxene + garnet = melt + orthopyroxene. This means that orthopyroxene will be crystallised during mantle melting or resorbed during magma crystallisation. While melting/crystallisation proceeds by this reaction all five phases are in equilibrium. The reaction is a statement of equilibrium. It is therefore incorrect to say “...the fractionated magma should be out of equilibrium with orthopyroxene”. I therefore think the next sentence does not follow: “This is consistent with the disappearance of orthopyroxene in our lower-temperature experiments...”. I also struggle to see how these inferences can be applied to a “fractionated magma” when the experiments are simulating batch melting without fractionation.

3. Lines 199-202. The text suggests that garnet crystallised from the magma will then react with

the surrounding peridotite. I struggle to see how this process could work in practice, other than on a very small scale. The peridotite and garnet must be in contact for the reaction to occur, and this will presumably be limited by the diffusion length scale, so would occur only on a small mass of material (not the 10% required). Then if this did occur, how do the trace elements get out from the solid reactants? Since this is a novel process, I think more justification of the feasibility is required. I therefore doubt that the model is consistent with the trace element constraints.

Reviewer #3 (Remarks to the Author):

Thanks for the thorough replies to my comments including reference citations. The replies are well written and reasoned. I think the authors did their considerable efforts.

These comments and associated changes to the manuscript make the paper suitable for publication in my opinion.

Like the Sobolev et al. 2005 paper, I disagree with the model but I am sure this paper will stimulate discussion.

Point-by-point response file

Response to Reviewer #1

Reviewer #1:

- 1) Lines 10-13: This sentence is a bit misleading and that is because melt with relatively high MgO are considered near to primary melts whereas melts with low MgO are evolved melts after olivine crystallization (at low pressure). Here authors need to better specify.

Reply: The sentence has been fixed.

- 2) Line 20: Here I suggest to specify the depth (pressure).

Reply: The sentence has been fixed.

- 3) Line 21: Specify the depth (pressure).

Reply: The sentence has been fixed.

- 4) Line 24: How do you concile the isotopic signature and your major and trace element model?

Reply: This is an important issue, but it is beyond the scope of this paper. Honestly, we can't answer this question now. We will focus on this issue in the future studies.

- 5) Line 28: But it should also match the isotopic signature! What about it?

Reply: Please see the reply under the question NO. 4.

- 6) Line 32: I found only experimental data at 3 GPa in the paper, why do you refer to this P range?

Reply: This P range is related to experimental results of partial melting of peridotite (Walter, 1998) rather than the experiments of this paper.

Walter, M. J. Melting of Garnet Peridotite and the Origin of Komatiite and Depleted Lithosphere. *Journal of Petrology* 39, 29-60 (1998).

7) Line 72: Supplementary material also should contain a brief discussion about the attainment of chemical equilibrium in experimental charges. That is requested because experimental durations are not so long (around 24 hrs).

Reply: A brief discussion has been added.

8) Lin 78: It is surprising that MgO does not follow a regular decreasing with crystallization extent. Did you evaluate any Fe loss in high-T experiments?

Reply: Yes, this result also surprised us. We have evaluated the Fe loss by mass balance calculation. The results show that there are negligible Fe loss in experiments.

9) Line 95: I suggest to refer to one depth (or pressure) here.

Reply: The sentence has been fixed.

10) Line 103: Here would be very nice to calculate modal and chemical features of garnet pyroxenites produced by experiments (solid phases at varying T) to be compared to Hawaiian xenoliths.

Reply: The Hawaiian garnet pyroxenite xenoliths were hosted by the rejuvenation stage magma. It is highly possible that the chemical exchange between them have changed the composition of the former. Therefore, we do not make such comparison in this paper.

11) Line 107: Here I do not understand why after garnet and cpx high P segregation the resulting evolved melt still reacts with peridotite opx. The reactivity of melts is controlled by the silica activity; how much is it in the experimentally-derived melts?

Reply: Peridotite-derived primary magma is formed by reaction of $Cpx + Grt + Ol = Melt + Opx$ (Walter, 1998). Reverse reaction should occur during melt crystallization. Therefore, the fractionated magma should be out of equilibrium with Opx and react with the latter to form $Cpx + Grt + Ol$.

12) Line 112: Does it mean that melts are rising within magmas conduits? Please specify.

Reply: Please see the reply under NO. 14.

13) Line 114: This is true only if magmas rises through conduits or fractures, is it the case?

Reply: Please see the reply under NO. 14.

14) Line 131: Yes of course. Anyway, the opx dissolution occurs through reactive porous flow. According with your model, after grt+cpx segregation at about 90 km the evolved melt migrate within conduits or fracture (if possible?) and at shallower levels (about 1.4 GPa) they migrate by reactive porous flow? Is it correct? This is an important aspect of the model the authors should better specify.

Reply: Yes, the melt migrates within conduits after Grt+Cpx segregation and at shallower levels they migrate and react with Opx by reactive porous flow. It has been specified in revised manuscript.

15) Line 157: In between these two steps there are about 45 km (about 15 kbar) of lithospheric mantle, how do the melts rise within any chemical modification?

Reply: According to your tips, we have specified the rising processes. Please see the reply under NO. 14. The depth of the first step is about 90 km and that of the second step is about 60 km. We believed that magma migrated through conduits between these two depths.

16) Line 191: Why do you select this composition?

Reply: We think that the major and trace element composition should match with each other. Ti, as a moderately incompatible element, has well-characterized behavior during peridotite melting (Walter, 1998; Prytulak & Elliott, 2007). Our starting material (P19) is similar in composition with partial melts of mantle peridotite at 3.0 GPa with a melt fraction about 24% (See methods). Since the TiO₂ concentration in P19 is 0.8 wt%, therefore the corresponding source peridotite should have about 0.2 wt% TiO₂ according to Walter's experimental results (Walter, 1998). The TiO₂ concentration in primitive mantle peridotite and depleted mantle peridotite should be about 0.22 wt% and 0.12 wt% (Prytulak & Elliott, 2007), respectively. So we assumed that the composition of source peridotite is similar to a 75:25 mixture of primitive mantle and depleted mantle.

Walter, M. J. Melting of Garnet Peridotite and the Origin of Komatiite and Depleted Lithosphere. *Journal of Petrology* 39, 29-60 (1998).

Prytulak, J. & Elliott, T. TiO₂ enrichment in ocean island basalts. *Earth Planet. Sci. Lett.* **263**, 388-403 (2007).

17) Line 203: This sentence is not clear.

Reply: The sentence has been fixed.

18) Line 208: I do not see the melts after the second step of the model here proposed in the suppl. fig 1. Maybe here they want to refer to Fig 1?

Reply: This refers to the melts formed at the second step of the model, that is the melt after Cpx and Grt crystallization.

19) Line 225: Not clear How do you estimate melt fraction in the plume. If few pyroxenite are in the plume they strongly enlarge the melt productivity.

Reply: Here, we mean the melt fraction of the lherzolite instead of the whole plume, which contains lherzolite and small proportion of pyroxenite.

20) Line 226: I do not understand this conclusion. Why the melts need a high degree of fractionation?

Reply: As we discussed in the introduction (Line 34-37), the enrichment in incompatible elements of Hawaiian primary magmas is consistent with low degrees (e.g. ~ 5 wt%) of melting of lherzolite. If the plume contains some proportion of pyroxenite, then its temperature must be high enough to compensate the negative buoyancy arose by the pyroxenite. Then the melt fraction must be high, which is conflict with the trace element abundance characteristics of Hawaiian lavas. In this case, the primary magmas also need to experience a high degree of fractionation.

21) Line 227: I understand that authors in this paragraphs want to exclude an important role of pyroxenite in the Hawaiian plume; however, I suggest to delete or rewrite the last two sentences because they are misleading.

Reply: Here, we want to tell that our model can not entirely rule out the existence of pyroxenite in the plume source region. However, the proportion of pyroxenite should be limited to smaller than 8 wt% and the primary magma must have experienced a high degree of fractionation in the deep magma chamber.

22) Line 229: If magma rises by very low migration rate it is expected to re-equilibrate with surrounding mantle lithosphere, why this does not occur?

Reply: The re-equilibration reaction between the evolved primary magma and the orthopyroxene in surrounding peridotite almost stops soon after the formation of a thin layer consisting of olivine, clinopyroxene and garnet that isolate the melts from peridotite.

23) Line 236: Yes but it strongly depends on the pressure and MgO of melt; I suggest to specify here. That is because at pressure of 2 GPa many basalts have cpx on its liquidus and thus they crystallize cpx.

Reply: The sentence has been fixed.

24) Line 272: I strongly suggest to insert here some issues regarding the approach of equilibrium and Fe-loss estimates in experiments.

Reply: The approach of equilibrium and Fe-loss estimates have been inserted in revised manuscript.

Response to Reviewer #2

Reviewer #2:

- 1) Lines 74-75. Reiterating from my previous review that saying “orthopyroxene disappeared” when describing the experiments implies that there was orthopyroxene in the starting material, or it crystallised and then was resorbed. The last sentence in the paragraph says the starting material was “homogeneous hydrous glass”, so it cannot be the first, and since the phase proportions were determined following quenching, it cannot be the second. The response to this comment in my first review (Reviewer 2, point 4 in the rebuttal) did not address this.

Reply: If we only discussed one single lower temperature experiment (e.g. the Run R1243 with temperature of 1300 °C), then your description is correct: the Opx is unstable, it never appears in the sample and there is no reaction. However, here we want to describe the whole crystallizing process from 1400 °C to 1200 °C based on our experimental results. Let us consider a contentious temperature decreasing process of our starting materials at 3 GPa, the Opx should first appears at 1400 °C and then disappears at temperatures lower than 1350 °C. In this case, there is no problem to say that the disappearance of Opx should attribute to the reaction.

- 2) Lines 103-109. The text cites experimental literature that shows in the pressure range under consideration mantle melting proceeds by the reaction olivine + clinopyroxene + garnet = melt + orthopyroxene. This means that orthopyroxene will be crystallised during mantle melting or resorbed during magma crystallisation. While melting/crystallisation proceeds by this reaction all five phases are in equilibrium. The reaction is a statement of equilibrium. It is therefore incorrect to say “...the fractionated magma should be out of equilibrium with orthopyroxene”. I therefore think the next sentence does not follow: “This is consistent with the disappearance of orthopyroxene in our lower-temperature experiments...”. I also struggle to see how these inferences can be applied to a “fractionated magma” when the experiments are simulating batch melting without fractionation.

Reply: In the source region, with temperature decreasing the reaction should be $\text{Melt1} + \text{Opx} = \text{Melt2} + \text{Cpx} + \text{Grt} + \text{Ol}$. In this case the Melt2 should be equilibrated with Opx. However, when the magma was extracted to form the source region to the deep

magma chamber, with temperature decreasing the reaction should be mainly occurred as $\text{Melt1} = \text{Melt3} + \text{Cpx} + \text{Grt}$ due to absence of Opx according to our experimental results. In this case, the Melt3 should be out of equilibrium with orthopyroxene. In the magma chamber, the precipitating Cpx and Grt should be kept the bottom side. So, we believed that the crystallizing of primary magma in deep chamber should be closed to batch crystallization. The real “fractionation” occurs after the magma rising from the chamber.

- 3) Lines 199-202. The text suggests that garnet crystallised from the magma will then react with the surrounding peridotite. I struggle to see how this process could work in practice, other than on a very small scale. The peridotite and garnet must be contact for the reaction to occur, and this will presumably be limited by the diffusion length scale, so would occur only on a small mass of material (not the 10% required). Then if this did occur, how do the trace elements get out from the solid reactants? Since this is a novel process, I think more justification of the feasibility is required. I therefore doubt that the model is consistent with the trace element constraints.

Reply: Here, the 10 % is the ratio of the broken garnet to the total crystallized garnet. This number is changed to 3%, if we calculate the ratio of the broken garnet to the total primary magma. We have no idea about how to evaluate the amount of garnet accurately. However, we know that olivine is the most abundant mineral in the surround mantle peridotite (e.g. ~ 70wt% for harzburgite), which definitely increased probability of the contaction between garnet and olivine. On the other hand, the rising magma may carry small amount of garnet from the deep chamber and these garnets will decompose at lower pressures. Actually, our model solutions still fit actual compositions of Hawaiian magmas with 20% relative error without including the decomposition of garnet (Fig. 3a). Here, we just want to make our model more closing to the actual process.